# HIV-Associated Insults Modulate ADAM10 and Its Regulator Sirtuin1 in an NMDA Receptor-Dependent Manner

**DOI:** 10.3390/cells11192962

**Published:** 2022-09-22

**Authors:** Claudia Lopez Lloreda, Sarah Chowdhury, Shivesh Ghura, Elena Alvarez-Periel, Kelly Jordan-Sciutto

**Affiliations:** 1Department of Neuroscience, University of Pennsylvania, Philadelphia, PA 19104, USA; 2Department of Oral Medicine, School of Dental Medicine, University of Pennsylvania, Philadelphia, PA 19104, USA; 3College of Arts and Sciences, University of Pennsylvania, Philadelphia, PA 19104, USA; 4Department of Pharmacology, University of Pennsylvania, Philadelphia, PA 19104, USA

**Keywords:** HIV, HIV-associated neurocognitive disorders, amyloid precursor protein processing, ADAM10, Sirtuin1

## Abstract

Neurologic deficits associated with human immunodeficiency virus (HIV) infection impact about 50% of persons with HIV (PWH). These disorders, termed HIV-associated neurocognitive disorders (HAND), possess neuropathologic similarities to Alzheimer’s disease (AD), including intra- and extracellular amyloid-beta (Aβ) peptide aggregates. Aβ peptide is produced through cleavage of the amyloid precursor protein (APP) by the beta secretase BACE1. However, this is precluded by cleavage of APP by the non-amyloidogenic alpha secretase, ADAM10. Previous studies have found that BACE1 expression was increased in the CNS of PWH with HAND as well as animal models of HAND. Further, BACE1 contributed to neurotoxicity. Yet in in vitro models, the role of ADAM10 and its potential regulatory mechanisms had not been examined. To address this, primary rat cortical neurons were treated with supernatants from HIV-infected human macrophages (HIV/MDMs). We found that HIV/MDMs decreased levels of both ADAM10 and Sirtuin1 (SIRT1), a regulator of ADAM10 that is implicated in aging and in AD. Both decreases were blocked with NMDA receptor antagonists, and treatment with NMDA was sufficient to induce reduction in ADAM10 and SIRT1 protein levels. Furthermore, decreases in SIRT1 protein levels were observed at an earlier time point than the decreases in ADAM10 protein levels, and the reduction in SIRT1 was reversed by proteasome inhibitor MG132. This study indicates that HIV-associated insults, particularly excitotoxicity, contribute to changes of APP secretases by downregulating levels of ADAM10 and its regulator.

## 1. Introduction

Human immunodeficiency virus (HIV)-associated neurocognitive disorders (HAND) are a spectrum of neurologic disorders characterized by cognitive, motor, and behavioral symptoms. In the antiretroviral therapy (ART) era, the milder forms of cognitive impairment are more prevalent compared to preART era, but HAND is still seen in approximately 50% of persons with HIV (PWH) [1,2,3]. HIV can effectively cross the blood–brain barrier (BBB); however, it does not directly infect neurons. Rather, HIV infects brain-resident macrophages, microglia, and astrocytes, which then unleash detrimental products that lead to neuronal and synaptic injury. This includes cytokines and glutamate, the latter inducing excitotoxicity via activation of NMDA receptors [4,5]. Other cells of the blood–brain barrier (BBB) such as pericytes can also be infected and contribute to BBB breakdown and cognitive impairment [6,7]. Additionally, antiretroviral therapies used to control HIV viral replication can also induce neuronal damage [8,9,10,11,12].

At the neuropathological level, HAND shares features found in other neurodegenerative diseases, such as Alzheimer’s disease (AD), including amyloid-beta (Aβ) protein aggregation [13]. Extracellular amyloid plaques, one of two of the pathologic hallmarks of AD, are large, insoluble protein aggregates made up of Aβ. Studies from our lab and others have found that Aβ is also present in tissue of postmortem HAND brain specimens and in the CSF of patients with HAND [14,15,16,17,18,19]. However, in contrast to AD, Aβ aggregates in HAND appear as intracellular diffuse oligomers rather than large insoluble plaques [18,20,21]. Various processes, many of which are seen in HAND, are associated with increased Aβ production, including oxidative stress, excitotoxicity, endoplasmic reticulum stress, and neuroinflammation [22]. HIV and HAND in vitro as well as in vivo models also support the idea that Aβ is an important neuropathological marker—Aβ is increased in disease-relevant regions in HIV transgenic rats and in simian immunodeficiency virus-infected rhesus macaques [23,24,25]. HIV viral proteins, in this case Tat, can inhibit Aβ-degrading enzymes such as neprilysin, leading to Aβ accumulation [26].

Molecularly, these aggregates are formed by the cleavage of amyloid precursor protein (APP) by β-secretases and subsequent cleavage by γ-secretase, which releases Aβ monomers that can later aggregate. The main β-secretase is β-secretase APP-cleaving enzyme 1 (BACE1). APP is also processed in the non-amyloidogenic pathway, in which APP is cleaved by an α-secretase, mainly A Disintegrin and metalloproteinase domain-containing protein 10 (ADAM10), which precludes Aβ generation and releases the neuroprotective molecule sAPPα [27,28,29]. Changing levels of amyloidogenic versus non-amyloidogenic enzymes that cleave APP can shift toward the generation of greater quantities of Aβ over time. In at least two pathologic conditions, AD and HAND, BACE1 protein levels are significantly increased in the disease state [18,30,31]. In an in vitro model of HIV-mediated neurotoxicity, BACE1 protein levels are upregulated and mediate neuronal damage through an APP-dependent mechanism [18]. Additionally, viral proteins such as Tat and ART drugs also increase levels of BACE1 [24,32,33]. As for the non-amyloidogenic pathway, ADAM10 levels are decreased in AD, in an AD mouse model that contains disease-associated mutations in APP, and in primary neurons treated with Aβ [34,35], possibly leading to decreased non-amyloidogenic processing of APP, as evidenced by decreased levels of sAPPα in the CSF of patients with AD [36]. We and others have also shown that excitotoxic injury by NMDA or glutamate similarly reduces levels of the ADAM10 and non-amyloidogenic processing in primary neurons [18,37,38]. Therefore, it is crucial to understand how APP secretase levels and their regulators change in response to neurotoxic stimuli, particularly in the context of HAND.

ADAM10 levels have been shown to be regulated by Sirtuin-1 (SIRT1), an NAD+ dependent deacetylase implicated in aging and neurodegeneration [39,40]. Importantly, SIRT1 plays a role in regulating APP processing; specifically, previous studies have described increased SIRT1 in response to caloric restriction, which mediated a decrease in Aβ in the brains of an AD mouse model [41]. Caloric restriction also resulted in decreased levels of transcription factors that inhibit non-amyloidogenic processing of APP by α-secretases [41,42]. It was also found that viral SIRT1 overexpression increased sAPPα while overexpression of a dominant-negative SIRT1 construct resulted in elevations of Aβ and a reduction in sAPPα, suggesting that modulation of Aβ levels may be mediated by changes in SIRT1 levels [41]. This may be due to the modulation of ADAM10 by SIRT1 [43].

SIRT1 is decreased in aged neurons in rodents as well as in affected brain regions such as the frontal cortex in AD patients [44,45]. It has also been proposed as a crucial player in HIV pathology [46]. SIRT1 mRNA levels are decreased in microglia and macrophages isolated from macaques infected with simian immunodeficiency virus that develop encephalitis (SIVE) [47]. Similarly, SIRT1 protein levels are decreased in HIV transgenic rats, which decrease further with age [48,49]. Using a triculture model, in which human-induced pluripotent stem cell-derived neurons and astrocytes were incubated with HIV-infected microglia, Sirtuin signaling was significantly decreased in neurons in the context of HIV-infected microglia [50]. Taken together, these findings suggest that APP processing may shift toward amyloidogenic processing by concomitant increases in the protein levels of APP secretases BACE1 and decreases in ADAM10, which may be regulated by Sirtuin levels and signaling. Thus, in this study, we sought to examine whether HIV-associated insults altered protein levels of the non-amyloidogenic enzyme ADAM10 and its regulator SIRT1.

## 2. Materials and Methods

### 2.1. Dissection and Maintenance Primary Rat Neuronal Cultures

All experiments were performed in accordance with the guidelines set forth by the University of Pennsylvania Institutional Animal Care and Use Committees. Primary rat cortical cultures were prepared from embryonic day 18 Sprague–Dawley CD rat embryos (Charles River Laboratories, Wilmington, MA, USA), as previously described [18]. Briefly, brains were isolated, and cortices dissected and incubated for 25 min in HBSS containing 2.5% Trypsin (Invitrogen, Waltham, MA, USA) and 80 U/mL DNase (Sigma-Aldrich, St. Louis, MO, USA) at 37 °C. Cortices were then rinsed twice with HBSS, mechanically disaggregated in DMEM + 10% FBS, passed through a 70 μm strainer and resuspended in neurobasal medium supplemented with B27 and GlutaMAX (all from Thermo Fisher Scientific, Waltham, MA, USA). Cells were plated on poly-L-lysine (Peptides International, Inc., Louisville, KY, USA) coated 6-well or 24-well plates (CC7682-7506 and CC7682-7524, USA Scientific, Ocala, FL, USA) at a concentration of 300,000–500,000 cells/mL. At day in vitro (DIV) 2–3, cells were treated with the anti-mitotic agent arabinosylcytosine C (AraC) (2 μM; C6645, Sigma-Aldrich, St. Louis, MO, USA) to inhibit proliferation of astrocytes and obtain pure neuronal cultures (99.9%). Cultures were maintained in neurobasal media with B27 supplement at 37 °C with 5% CO_2_. On DIV10 and DIV16, 50% of media were removed and replaced with fresh media.

### 2.2. HIV/MDM Generation

All human peripheral blood mononuclear cells, including monocytes, were obtained from the University of Pennsylvania Center for AIDS Research Virology Core. Monocytes from healthy human donors were plated onto 6-well Cell-Bind plates in DMEM with FBS, penicillin/streptomycin, and nonessential amino acids. Monocytes were then differentiated into macrophages over the course of 7 days with granulocyte–macrophage colony-stimulating factor (GM-CSF) treatments at DIV 1 and 3 with a 100% media change at DIV 6. GM-CSF drives macrophages into a proinflammatory state, reflecting the macrophage profile in HAND [51]. HIV Jago stocks were prepared in primary T-lymphocytes derived from healthy volunteer donors through the University of Pennsylvania Center for AIDS Research Virology Core. HIV Jago is a macrophage-tropic strain that was isolated from the CSF of a patient with HIV dementia and was used in this study for its ability to infect macrophages [52]. After complete differentiation around DIV 7, macrophages were incubated with 50 ng for every 10^6^ cells of HIV Jago strain for 24 h. After 24 h, the media were removed, and cells were washed with DMEM media three times. After establishing infection, macrophages were evaluated for productive infection by looking for formation of syncytia, which indicates cell fusion into large multinucleated formations. Supernatants were collected on DIV 3, 6, 9, 12, and 15. HIV p24 levels were measured using a p24 ELISA to confirm productive infection. HIV/MDM supernatants were kept at −80 °C until treatment.

### 2.3. Neuronal Culture Treatments

Neuronal cultures were maintained until DIV 21. At DIV21-22, neurons were treated with Mock/MDMs or HIV/MDMs. Cells were treated with the same supernatants from the same donor (ND527) unless otherwise stated. For experiments with different sets from different donors, we titrated every supernatant set to 50% neurotoxicity to ensure that we were eliciting the same toxic insult. To mimic excitotoxicity, cells were treated with NMDA (Tocris, Bioscience, Bristol, United Kingdom, 0114) or vehicle (H_2_O). The dilutions used for each set of HIV/MDM supernatants was determined by titrating to 50% neurotoxicity as quantified by immunohistochemistry for MAP2. Chloroquine (CQN; 1 or 2 μM; C6628, Sigma Aldrich, St. Louis, MO, USA) or vehicle (DMSO; 276855, Millipore Sigma, Burlington, MA, USA) treatment was performed 4 h prior to treatment with Mock/MDMs, HIV/MDMs, or NMDA (5, 10, 15 and 20 μM). Treatment with MG132 (1 or 5 μM; 10012628, Cayman Chemical, Ann Arbor, MI, USA), MK-801 (1 μM; 0924, Tocris Bioscience, Bristol, United Kingdom) and their respective vehicles (DMSO and H_2_O) was performed 1 h prior to treatment.

### 2.4. Immunoblotting

Protein levels were analyzed via Western blotting of cell lysates collected at the timepoint determined after treatment. Briefly, after treatment, media were removed, and cells were washed three times with cold PBS. Cells were lysed using whole cell lysis buffer (WCLB) consisting of 50 mM Tris pH 7.5, 120 mM NaCl, 0.5% NP-40, 0.4 mM NaF, 0.4 mM Na3VO4, and Halt™ Protease and Phosphatase Inhibitor Cocktail (Thermo Fisher Scientific, Thermo Fisher Scientific, Waltham, MA, USA). Using 100 µL of WCLB, cells were scraped mechanically, and protein samples were collected. Samples were then centrifugated for 10 min at 4 °C at 20,000× *g*, and supernatants were collected and kept at −80 °C. Protein concentrations were determined using Bradford assay. Samples were prepared with reducing agent, LDS loading buffer, and water, and heating to 70 °C. Equal amounts of protein were then loaded onto 4–12% Bis-Tris NuPAGE gradient cells (Thermo Fisher Scientific, Waltham, MA, USA) and separated. Proteins were transferred onto Immun-Blot polyvinylidene difluoride membrane (PVDF; Bio-Rad, Hercules, CA, USA). Membranes were blocked for 30 min with 5% bovine serum albumin (BSA) in Tris-buffered saline 0.1% tween-20 (TBS-T) at room temperature. Next, membranes were incubated in primary antibodies overnight at 4 °C. The primary antibodies used and their dilutions were: ADAM10 (ab1997, 1:1000; Abcam, Cambridge, United Kingdom), SIRT1 (ab110304, 1:500; Abcam, Cambridge, United Kingdom,), BACE1 (5606 s, 1:2000; Cell Signaling Technology, Danvers, MA), and β-actin (3700, 1:20,000; Cell Signaling Technology, Danvers, MA, USA). Membranes were then taken out of primary antibody, washed with TBS-T three times, and incubated in horseradish peroxidase (HRP)-conjugated secondary antibody diluted in TBS-T for 30 min at room temperature. The secondary antibodies used were Goat anti-Rabbit IgG (H + L) Secondary Antibody and Goat anti-Mouse IgG (H + L) Secondary Antibody (PI31460 and PI31444, 1:5000; Thermo Fisher Scientific, Waltham, MA, USA). Finally, membranes were incubated with Luminata Classico electrochemiluminescence (ECL) for 3 min to visualize proteins (WBLUC0500; Millipore Sigma, Burlington, MA). Images were captured using ChemiDoc Touch imaging system (Bio-Rad Laboratories, Hercules, CA, USA). Densitometric analysis of band intensities was conducted using Image Lab software version 6.1.0 (Bio-Rad Laboratories, Hercules, CA, USA). Protein levels were quantified for each sample and normalized to β-actin that functioned as a loading control. Protein levels of experimental groups were normalized to levels of the untreated sample. In addition, the presence of ADAM10 and SIRT1 mRNA in rat neuronal cortical cultures was confirmed through reverse transcription–polymerase chain reaction (RT-PCR) (Appendix A).

### 2.5. Immunohistochemistry

To assess neurotoxicity, cells were plated in 24-well plates and treated with HIV/MDM supernatants for 24 h at DIV 21. After treatment, media were removed, and cells were washed twice with phosphate-buffed saline (PBS) and fixed for 20 min with 4% paraformaldehyde. Paraformaldehyde was removed, and cells were washed twice with PBS and three times with PBS with 0.1% Tween 20 (1706531, Bio-Rad Laboratories, Hercules, CA, USA) (PBS-T). Next, cells were blocked and permeabilized with 0.2% BSA and 0.1% Triton X-100 (9036-19-5, Millipore Sigma, Burlington, MA, USA) in PBS for 30 min. Cells were then washed again three times with PBS-T and left to incubate with anti-MAP2 antibody (4542S, 1:3000, Cell Signaling Technology, Danvers, MA, USA) in normal antibody diluent overnight at 4 °C. After overnight incubation, MAP2 primary antibody was washed off with three PBS-T washes. Cells were incubated with goat anti-mouse IgG secondary antibody conjugated to fluorescein isothiocyanate (FITC; 1:200) and DAPI (1:4000) diluted in normal antibody diluent for 30 min. After three more washes with PBS-T, cells were left in PBS. Cells were imaged in Keyence BZ-X-700 digital fluorescent microscope (Osaka, Japan). Images were captured at 20× at 25 points per well with 2–3 wells per treatment. BZ-X Keyence software version 1.3.0.3 was used to quantify the number of neurons by identifying the number of MAP2+/DAPI+ cells and averaging this number for each treatment condition.

### 2.6. Statistical Analysis

Each treatment group was normalized to the untreated group (represented by dotted lines in the graphs). Paired *t* tests were performed for experiments with two conditions and repeated-measures ANOVA were performed for experiments with three conditions or more followed by Tukey’s post hoc test if significant. For experiments tested with ANOVA, normality was tested and confirmed using the Shapiro–Wilk test. Significance was set at *p* < 0.05. Correlation analysis was conducted using Spearman’s rank correlation coefficient. Data were analyzed using GraphPad Prism statistical software version 9.4.0 (San Diego, CA, USA), and data are expressed as mean fold change from UT ± SEM.

## 3. Results

### 3.1. HIV/MDM Supernatants Lead to Neurotoxicity in Rat Neuronal Cultures

To model the effects of HIV infection on neurons, we treated primary rat cortical cultures with supernatants collected from HIV-infected monocyte-derived macrophages (HIV/MDMs) as previously described [53,54,55]. To account for the variability of human macrophages and their immune response, supernatants were examined across a range of dilutions, and the concentration in which Mock/MDM supernatants were not toxic and HIV/MDM supernatants resulted in a 50% reduction in neurons was used for further experimentation. For supernatants derived from one human monocyte donor, we show that a dilution of 1:300 led to a 50% reduction in viable MAP2-positive neurons (Figure 1A,B).

### 3.2. HIV/MDMs Reduce Levels of ADAM10 Protein

Previous studies from our lab showed that HIV/MDMs increased levels of the amyloidogenic APP secretase BACE1 and that this upregulation contributed to neurotoxicity via APP [18]. To gain insight into how other APP secretases changed in response to HIV-associated insults, we assessed expression levels of the non-amyloidogenic secretase ADAM10 in response to HIV/MDMs. Treatment of five different preparations of primary purified rat neurons with the same set of HIV/MDMs significantly decreased protein levels of ADAM10 as compared to Mock infected supernatants derived from uninfected macrophages (Figure 2A,B). In each rat cell culture preparation, HIV/MDMs decreased ADAM10 while Mock/MDM did not. However, it is well known that human macrophages respond differently to HIV infection, and we have previously observed variability between the monocyte-derived macrophages and the level of toxicity of the supernatants. Therefore, we wanted to test whether the ADAM10 decrease was observed across different human donors in the same preparation of rat neuronal culture. For this, we treated the same rat neurons with supernatants from five different macrophage donors (each pair represents the uninfected macrophages and infected macrophages for each donor). We found that HIV/MDMs consistently decreased protein levels of ADAM10 across five different macrophage donors in the same preparation of rat neuronal cultures (Figure 2C,D). In this case, for each macrophage donor, HIV/MDMs decreased ADAM10 while Mock/MDMs did not. These results indicate that, in addition to upregulating the amyloidogenic BACE1 enzyme, treatment with HIV/MDM supernatants also downregulates levels of the non-amyloidogenic secretase ADAM10 in primary neurons.

### 3.3. NMDA Receptor Activation Is Necessary and Sufficient for ADAM10 Decrease by HIV/MDMs

Our studies and others have shown that the neurotoxicity of HIV/MDM supernatants is dependent on NMDA receptors, since NMDA receptor antagonists such as MK-801 can block neuronal cell death induced by HIV/MDMs [53]. Similarly, in our cell cultures, MK-801 blocked toxicity by the HIV/MDMs used in this study. Furthermore, Stern et al., in 2018, showed that the induction of BACE1 by HIV/MDMs was blocked by NMDA receptors antagonists MK-801 and AP-5, suggesting that the modulation of APP secretase levels is dependent on activation of these receptors. To determine whether NMDA receptor activation mediated the decrease in ADAM10 by HIV/MDMs, we pretreated neurons with NMDA receptor antagonists for 1 h prior to treatment with HIV/MDMs. We found that blocking NMDA receptors with MK-801 rescued levels of ADAM10 back to their untreated levels and were significantly different from cells only treated with HIV/MDMs (Figure 3A,B). Based on these results, cells were treated with NMDA to examine if NMDA receptor activation was sufficient to induce APP secretase changes. We performed a dose-curve experiment in which we tested four different concentrations of NMDA (5, 10, 15, and 20 μM) for 16 h. The highest dose of NMDA (20 μM) led to a significant decrease in ADAM10 protein levels when compared to treatment with vehicle (Figure 3C,D).

### 3.4. HIV/MDMs Reduce Levels of the ADAM10 Regulator SIRT1

SIRT1 has been shown to regulate levels of non-amyloidogenic processing of APP [41,56]. For this reason, we examined whether HIV/MDM supernatants might also alter ADAM10 regulator, SIRT1. Similar to ADAM10, treatment with HIV/MDM supernatants led to a decrease in SIRT1 protein levels (Figure 4A,B). As described before, to examine whether this was reproducible, we looked at four additional human macrophage donors. HIV/MDMs from all five macrophage donors decreased levels of SIRT1 when compared to Mock/MDMs (Figure 4C,D).

### 3.5. NMDA Receptor Activation Is Necessary and Sufficient for SIRT1 Decrease by HIV/MDMs

To examine whether SIRT1 decrease by HIV/MDMs occurred through the same mechanism as the ADAM10 decrease, we looked at SIRT1 levels after blocking NMDA receptors with MK-801 prior to treating with HIV/MDMs. Again, we found that HIV/MDMs decreased levels of SIRT1 when compared to cells treated with Mock/MDMs and that this decrease was avoided by blocking NMDA receptors with MK-801 (Figure 5A,B). In addition, we also examined whether NMDA receptor activation led to changes in the Sirtuin pathway. We found that NMDA at the 20 μM concentration decreased levels of SIRT1, similar to the changes observed with ADAM10 (Figure 5C,D).

### 3.6. SIRT1 Decrease Precedes ADAM10 Decrease

To assess the mechanism of ADAM10 and SIRT1 downregulation in response to both HIV/MDMs and NMDA, we examined the levels of each protein at earlier timepoints with the purpose of determining whether ADAM10 and SIRT1 downregulation occurs simultaneously or if one precedes the other. First, we performed a time course that indicated that SIRT1 starts decreasing at 2 h while ADAM10 starts decreasing somewhere between 8 and 12 h (Appendix A). This led us to examine this early 2 h timepoint to further assess if SIRT1 was decreasing prior to ADAM10 and the possible mechanisms regulating this early decrease. At 2 h after treatment with NMDA, we observed a decrease in SIRT1, while ADAM10 levels remained the same (Figure 6A–C), suggesting that the SIRT1 decrease may occur prior to the ADAM10 decrease in response to NMDA and possibly HIV/MDMs. A correlation analysis of protein levels in HIV/MDMs-treated cells at 16 h revealed that protein levels of ADAM10 were correlated with protein levels of SIRT1 (Figure 6D), suggesting that there may be a mechanistic relationship between the two.

### 3.7. Autophagy Inhibitors Do Not Block Early SIRT1 Decrease by HIV/MDMs

Next, we sought to pinpoint the specific mechanism of the early downregulation of SIRT1 in response of HIV/MDMs and NMDA. We examined autophagy since SIRT1 downregulation had been shown to be mediated through autophagy in the context of senescence in fibroblasts [57]. To determine the mechanism of downregulation of SIRT1 at the 2 h timepoint, cells were treated with the lysosomal inhibitor chloroquine for 4 h prior to treatment with HIV/MDMs or NMDA. We used chloroquine doses found to increase LC3b, a readout of autophagy inhibition (Figure 7A,D,E,H). The doses used did not significantly increase the ratio of LC3bII to LC3bI; however, there was a trend toward an increase with chloroquine. Treatment with HIV/MDMs led to a decrease in SIRT1 while ADAM10 levels remained unchanged (Figure 7A,B). Our findings show that two different doses of chloroquine did not have an effect on the early SIRT1 decrease by HIV/MDMs (Figure 7A,B). As for the NMDA treatment, NMDA did not significantly decrease SIRT1 and did not change ADAM10 levels (Figure 7C,D). However, chloroquine actually further decreased SIRT1 when compared to cultures treated with vehicle, confirming the efficiency of the treatment and indicating that autophagy inhibitors did not rescue SIRT1 levels (Figure 7D).

### 3.8. Early SIRT1 Decrease in Response to HIV/MDM Supernatant Treatment Occurs via Proteasomal Degradation

Next, we examined the contribution of the proteasome pathway, as SIRT1 has been shown to be degraded by the proteasome in a Parkinson’s disease mouse model [58]. Cells were treated with the proteasome inhibitor MG132 for one hour prior to treatment with HIV/MDMs. Our results showed that the proteasome inhibitor MG132 blocked the downregulation of SIRT1 by HIV/MDM (Figure 8A,B). This suggests that SIRT1 reduction in response to HIV/MDMs is via the proteasome pathway.

## 4. Discussion

Our results have identified two mechanisms that contribute to the downregulation of ADAM10 and SIRT1 in the context of HIV-associated neurotoxic insults. At an early timepoint, we found that SIRT1 downregulation, which occurs prior to the downregulation of ADAM10, is dependent on the proteasome, pointing to a possible mechanism of protein degradation. At the later timepoint, we found that both ADAM10 and SIRT1 downregulation are dependent on glutamate receptors, specifically NMDA receptor activation. These findings parallel our previous study showing that BACE1 induction by HIV/MDMs is dependent on NMDA receptor activation [18]. Stern et al., in 2018, found that upregulation of BACE1 in our HIV/MDM model mediated neurotoxicity in an APP-dependent manner. This current study further illustrates that HIV-associated insults not only lead to increases in the pro-amyloidogenic secretase, BACE1, but also lead to decreases in the non-amyloidogenic secretase ADAM10, which would push processing of APP toward an amyloidogenic pathway. We also found that NMDA receptors mediate the reduction of ADAM10 caused by HIV/MDM supernatants. Additionally, NMDA receptor activation by itself was sufficient to induce a decrease in ADAM10. This finding is consistent with our previous study showing that NMDA and glutamate treatment of cortical neurons decrease ADAM10 [18].

However, certain studies have found the opposite: treatment of primary mouse cortical neurons with NMDA led to increased levels of ADAM10, α-secretase activity, and non-amyloidogenic processing of APP [59,60]. Possible reasons for this discrepancy are the concentration and duration of NMDA treatment impacting the type of receptors that are activated. Wan et al. treated cells with a high dose of NMDA (50 μM) for a short amount of time (5–90 min) while our study examines a more chronic timeframe [60]. Different concentrations of NMDA can play a role by selectively activating synaptic NMDA receptors, which are associated with neuroprotection, or extrasynaptic NMDA receptors, which are associated with calpain activation and cell death. Specifically, higher doses of NMDA increasingly activate more extrasynaptic NMDA receptors [61]. For example, another study showed that activation of extrasynaptic, but not synaptic, NMDA receptors increased Aβ production [38]. To this point, Hoey et al., in 2009, found that the effect they observed on increased non-amyloidogenic processing was mainly through synaptic NMDA receptors activation [59]. Therefore, we suspect that in our cell culture, we are activating extrasynaptic NMDA receptors that shift APP processing toward the amyloidogenic pathway. One future direction could be to selectively block extrasynaptic NMDA receptors to determine the effect this would have on HIV/MDM-mediated decreases in ADAM10 and SIRT1 and neurotoxicity.

Whether the ADAM10 decrease by either HIV/MDMs or NMDA leads to a change in APP processing remains to be studied. There is reason to think that a decrease in ADAM10 due to excitotoxicity might consequently shift APP processing, as shown by Lesné, in 2005, in which NMDA treatment led to decreased levels of sAPPα [37]. Other studies have similarly used NMDA receptor antagonists such as memantine, which preferentially blocks the extrasynaptic pool [62], to examine APP processing. For example, oral treatment with memantine for eight days in APP/presenilin1 (APP/PS1) transgenic mice, which develop Aβ plaques and memory deficits, decreased the soluble form of the Aβ peptide [63]. Similarly, memantine treatment also decreased Aβ in primary rat cortical neurons [63]. Oral administration of another NMDA receptor blocker RL-208 prevented the ADAM10 decrease seen in a mouse model of senescence [64]. A more recent study from the same group tested a new NMDA receptor antagonist named UB-ALT-EV, which also reduced Aβ deposition in an AD mouse model [65]. Overall, this research indicates a significant link between NMDA receptors and ADAM10. In addition to processing APP and possibly shifting the ratio of Aβ being generated, ADAM10 has other targets such as N-cadherin and regulates crucial processes such as cell–cell adhesion [66]. A proteomic analysis pinpointed 91 additional possible targets for neuronal ADAM10, including proteins involved in brain development [67]. Future studies should look into whether other ADAM10 targets besides APP are affected and whether these potentially regulate neuronal damage in HIV-associated insults.

As for the ADAM10 regulator SIRT1, previous studies examining SIRT1 in HIV and HAND had focused on microglia and macrophages, finding that SIRT1 was downregulated in these cells. These studies and others suggest that SIRT1 possibly functions to silence inflammatory genes in HIV [47,68,69,70]. Our study indicates that SIRT1 is decreased in response to HIV and excitotoxic insults, adding implications for neuronal SIRT1 in neuropathology, suggesting that HAND may be added to the range of neuropathological disorders characterized by decreased CNS SIRT1 levels [45,71]. Similar to ADAM10, SIRT1 decreases in response to NMDA receptor activation has been reported previously [72]. SIRT1 is a key regulator of many pathways; thus, this decrease may have many ramifications. Studies have linked a decrease in SIRT1 to increased amyloid plaques, uncontrolled neuroinflammation, and even tau pathology [68,73,74]. Future studies should address other consequences of SIRT1 downregulation in HAND, especially considering that the decrease is observed not only in macrophages and microglia in the brain, but also in neurons and astrocytes.

Importantly, we were able to identify the mechanism of downregulation for the early decrease in SIRT1. Two previous studies had identified autophagy and proteasomal degradation as possible mechanisms [57,58]. Our data do not support autophagy as the mediator of SIRT1 degradation in our HIV/MDM model, as the presence of an autophagy inhibitor does not reverse the reduction in SIRT1. However, the doses used did not consistently induce an increase in the ratio of LC3bII to LC3bI, which indicates effective inhibition of autophagic flux. Therefore, this pathway should not be entirely ruled out, and higher doses already known to increase this ratio in primary rat cortical neurons (such as 40 μM) should be tested in the future [75].

As for the ubiquitin–proteasome pathway, we found that MG132, a proteasome inhibitor, was able to rescue SIRT1 levels. A previous study found that SIRT1 is ubiquitinated by E3 ligase GRAIL/RNF128 and degraded by the proteasome [76]. Understanding the connection between NMDA receptor activation and the proteasome requires additional examination, as other studies have reported that excitotoxic stimulation of cultured hippocampal neurons decreases proteosome activity [77]. In rat hippocampal neurons, downregulation of NMDA receptors containing the subunit GluN2B decreased proteosome localization at the synapse [78]. Therefore, our observations that SIRT1 is degraded by the proteasome may be due to changing dynamics of the proteasome in response to NMDA receptor activation, the type of neuron being examined, the strength of the NMDA insult or the timing following NMDA stimulation. Studies specifically looking at HAND postmortem brain tissue found that there was an upregulation of immunoproteasome subunits in the frontal cortex, suggesting that there are other mechanisms in addition to glutamate excitotoxicity that are contributing to SIRT1 downregulation with HIV insults [79,80]. Together, these studies suggest that the link between the NMDA receptor and SIRT1 may also have distinct regulation depending on the extrasynaptic vs. synaptic NMDA receptor signaling akin to ADAM10, which warrants further investigation.

The specific regulatory mechanism of the ADAM10 decrease downstream of NMDA receptor activation was not identified, but our findings suggest that SIRT1 could be a possibility. Studies examining SIRT1 activators and inducers such as resveratrol show that they rescue levels of ADAM10 [35]. In one study, treating mouse neuroblastoma cells expressing APP with cilostazol and resveratrol resulted in Aβ reduction [81]. When the SIRT1 gene was silenced, cilostazol and resveratrol were not able to restore ADAM10 levels. Furthermore, overexpression of SIRT1 increased ADAM10 and sAPPα levels [81]. Cilostazol also increased RARβ levels, and inhibition of RARβ resulted in decreased ADAM10. These studies indicate that cilostazol increases ADAM10 levels through SIRT1 and RARβ. One possible mechanism suggested by this work is that SIRT1 deacetylates and activates RARβ, which then activates transcription of ADAM10. The importance of this signaling in the context of neurons and NMDA receptor stimulation should be considered further.

Importantly, it remains to be seen whether rescuing ADAM10 or SIRT1 can rescue HIV/MDM-mediated neurotoxicity. It is promising since many drugs that target these two pathways have been found to reduce amyloid plaque burden and rescue neurotoxicity in animal and cell culture models of AD. First, molecules that induce ADAM10, acitretin and all trans-retinoic acid, could be potential drugs to rescue ADAM10 levels in HIV and HAND models [82]. Other drugs have been found to provide neuroprotection through SIRT1. In vitro, resveratrol has been seen to provide neuroprotection through SIRT1 in SH-SY5Y and PC-12 cells treated with NMDA and Aβ [72,83,84]. This neuroprotection potentially translates to improved memory, as one study showed that APP/PS1 mice given resveratrol had improved spatial memory [35]. Another recent study found that betaine prevented memory deficits in mice injected with Aβ, a rescue than was blocked with the SIRT1 inhibitor sirtinol [85]. In patients with AD, various studies have found that resveratrol works to delay cognitive decline [86,87]. Our findings suggest that further examination of these compounds in the context of the mechanism proposed here is needed.

In light of the failure of BACE1 inhibitors to improve cognition in AD patients and the lack of effective adjunctive therapies for HAND, these SIRT1 and/or ADAM10 activators and inducers may present themselves as possible therapeutic alternatives [88,89,90,91,92]. Although a long way out from potential therapeutics, this study opens an avenue to examine the SIRT1-ADAM10 pathway as a pathway to protect neurons from damage in the context of insults associated with HAND.

## Figures and Tables

**Figure 1 cells-11-02962-f001:**
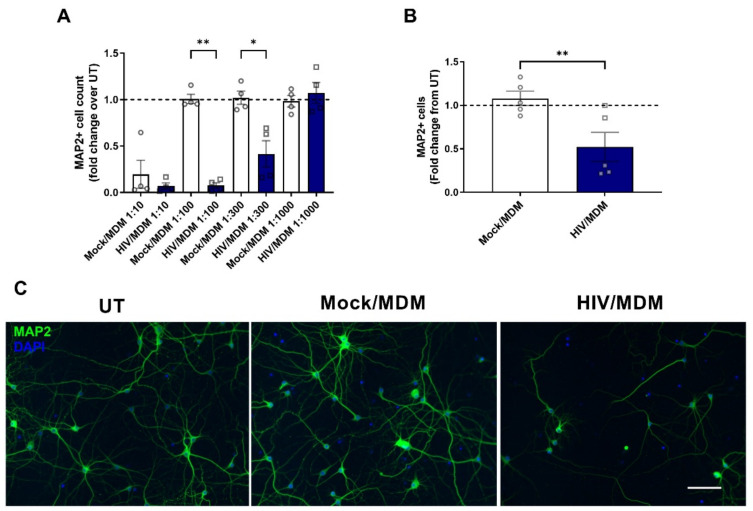
HIV/MDM supernatants treatment leads to neurotoxicity. Cells were fixed 24 h after treatment with Mock or HIV/MDMs. HIV/MDM supernatants derived from donor ND527 led to 50% neurotoxicity, as shown by MAP2-positive cell count (**A**,**B**). Statistical significance was calculated using one-way ANOVA (A) and paired *t* test (**B**). *** Represents *p* < 0.05, *** p* < 0.01 (*n* = 5). Dotted line represents untreated samples. Representative images are shown, with MAP2 in green and DAPI in blue. Scale bar = 100 μm (**C**).

**Figure 2 cells-11-02962-f002:**
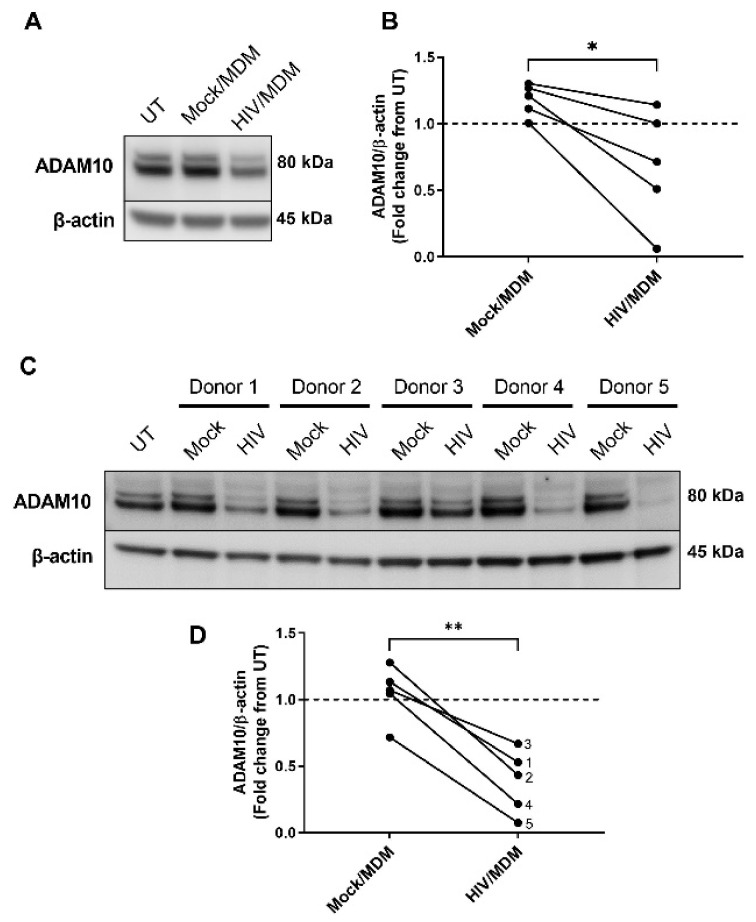
HIV/MDM supernatant treatment alters protein levels of APP secretase ADAM10. Protein lysates were collected from purified primary rat neuronal cultures after 16 h treatment with Mock or HIV/MDM supernatants. Representative blots are shown (**A**). Treatment with HIV/MDMs decreased levels of ADAM10 when compared to Mock/MDMs across five different isolates of primary rat cultures to account for variability across independent culture preparations (**B**). Statistical significance was calculated using paired *t* test. *** Represents *p* < 0.05 compared to Mock/MDMs (*n* = 5). Protein lysates were collected from rat neuronal cultures after 16 h treatment with five different sets of Mock and HIV/MDM supernatants to account for variability across individual macrophage donors. Representative blots are shown (**C**). Treatment with HIV/MDMs significantly decreased ADAM10 when compared to Mock/MDMs. Each number represents the macrophage donor pair (**D**). Statistical significance was calculated using paired *t* test. **** Represents *p* < 0.01 (*n* = 5). Dotted line represents untreated samples.

**Figure 3 cells-11-02962-f003:**
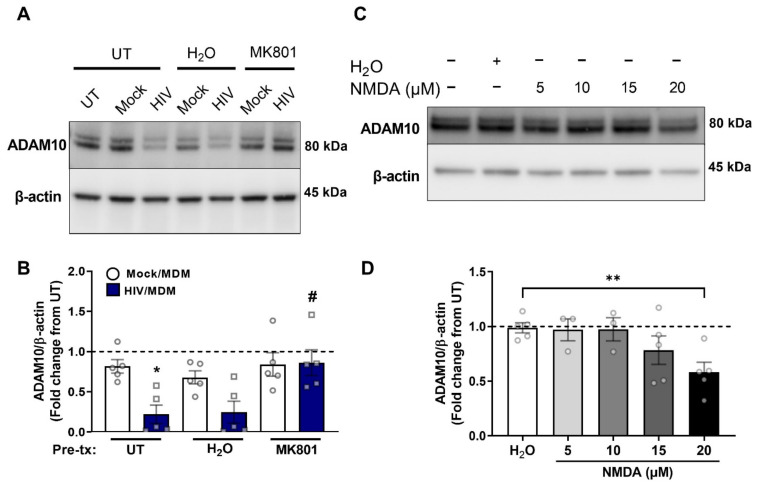
NMDA receptor activation is necessary and sufficient for ADAM10 decrease by HIV/MDMs. Protein lysates were collected from rat neuronal cultures after a 1 h pretreatment with vehicle (H_2_O) or NMDA receptor inhibitor MK-801 and a 16 h treatment with Mock or HIV/MDM supernatants. Representative blots are shown (**A**). Pretreatment with MK-801 seems to rescue ADAM10 decrease by HIV/MDMs (**B**). Statistical significance was calculated using a two-way ANOVA followed by Tukey’s post hoc. *** Represents *p* < 0.05 and *#* represents *p* < 0.05 compared to HIV/MDMs (*n* = 5). Protein lysates were collected from rat neuronal cultures after 16 h treatment with different concentrations of NMDA or vehicle (H_2_O). Representative blots are shown (**C**). The highest dose of NMDA, 20 μM, led to a significant decrease in ADAM10 (**D**). Statistical significance was calculated using one-way ANOVA. **** Represents *p* < 0.01 (*n* = 3–5). Dotted line represents untreated samples.

**Figure 4 cells-11-02962-f004:**
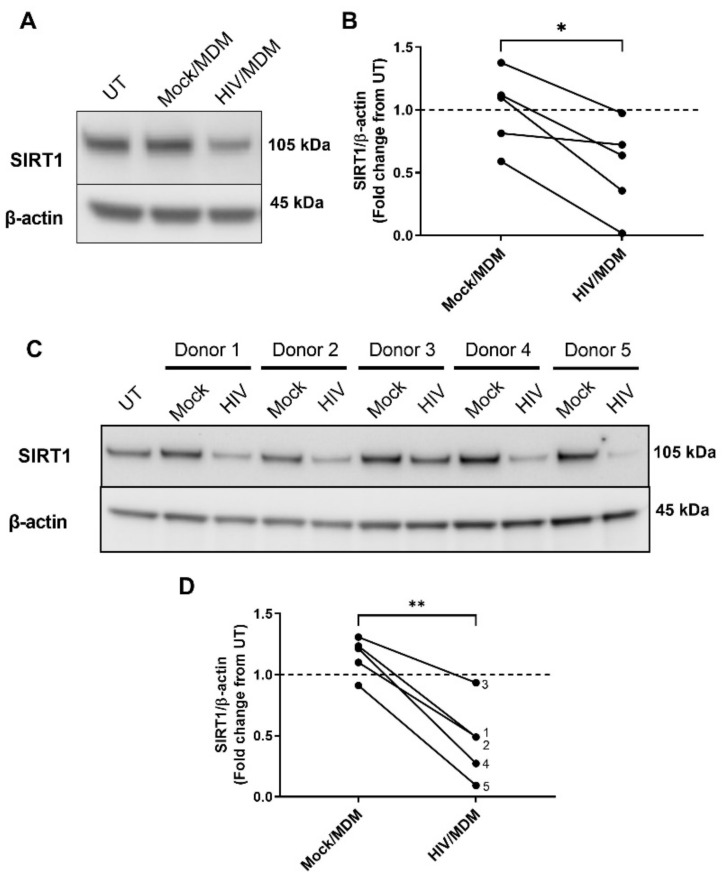
Protein lysates were collected from rat neuronal cultures after 16 h treatment with Mock or HIV/MDM supernatants. Representative blots are shown (**A**). Treatment with HIV/MDMs decreased levels of SIRT1 when compared to Mock/MDMs across five different preparations of primary rat cortical neurons (**B**). Statistical significance was calculated using paired *t* test. *** Represents *p* < 0.05 (*n* = 5). Protein lysates were collected from primary rat neuronal cultures after 16 h treatment with five different sets of Mock and HIV/MDM supernatants. Representative blots are shown (**C**). Treatment with HIV/MDMs significantly decreased SIRT1 when compared to Mock/MDMs. Each number represents the macrophage donor pair (**D**). Statistical significance was calculated using paired *t* test. *** Represents *p* < 0.05, ** *p* < 0.01 (*n* = 5). Dotted line represents untreated samples.

**Figure 5 cells-11-02962-f005:**
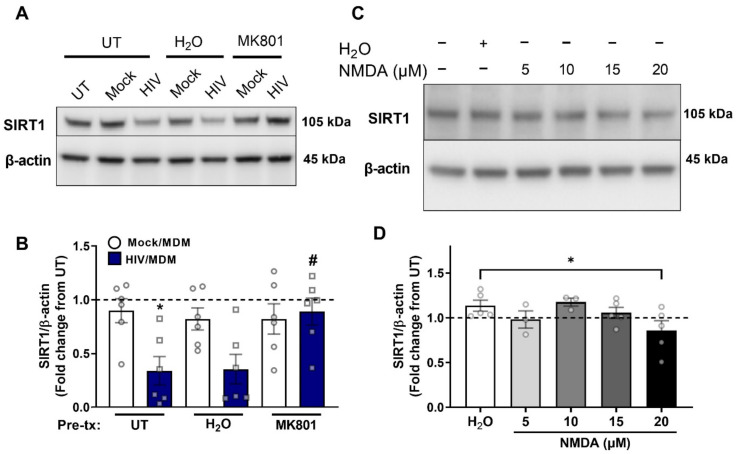
NMDA receptor activation is necessary and sufficient for SIRT1 decrease by HIV/MDMs. Protein lysates were collected from rat neuronal cultures after a 1 h pretreatment with vehicle (H_2_O) or MK-801 and a 16 h treatment with Mock or HIV/MDM supernatants. Representative blots are shown (**A**). Pretreatment with MK-801 rescues SIRT1 levels decreased by HIV/MDMs (**B**). Statistical significance was calculated using two-way ANOVA followed by Tukey’s post hoc. *** Represents *p* < 0.05 compared to Mock/MDMs and *#* represents *p* < 0.05 compared to HIV/MDMs (*n* = 6). Protein lysates were collected from rat neuronal cultures after 16 h treatment with different concentrations of NMDA or vehicle (H_2_O). Representative blots are shown (**C**). The highest dose of NMDA, 20 μM, led to a significant decrease in SIRT1 (**D**). Statistical significance was calculated using one-way ANOVA. *** Represents *p* < 0.05 (*n* = 3–5). Dotted line represents untreated samples.

**Figure 6 cells-11-02962-f006:**
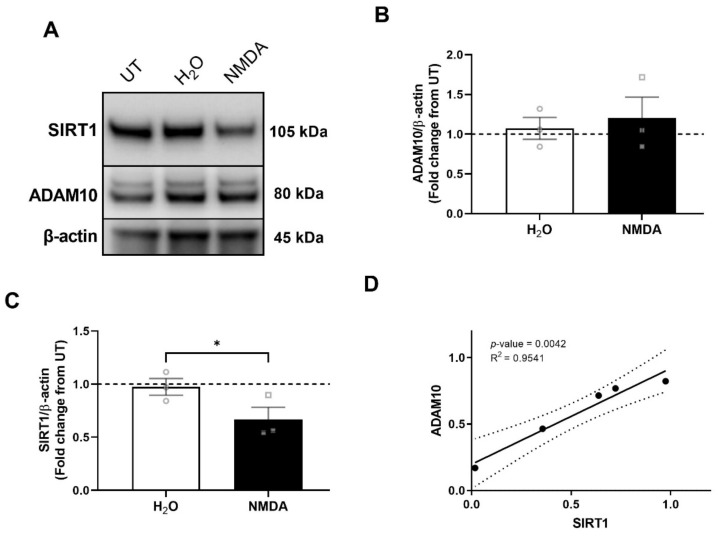
NMDA-mediated SIRT1 decrease occurs prior to ADAM10 decrease, and protein levels are correlated at a later timepoint. Protein lysates were collected from rat neuronal cultures after 2 h treatment with vehicle (H_2_O) or NMDA. Representative blots are shown (**A**). Treatment with NMDA did not alter levels of ADAM10 when compared to H_2_O (**B**). Treatment with NMDA at 2 h decreased levels of SIRT1 (**C**). Dotted line represents untreated samples. Statistical significance was calculated using paired *t* test. *** Represents *p* < 0.05 (*n* = 3). In a correlation analysis, levels of ADAM10 correlate with levels of SIRT1 in HIV/MDMs-treated samples at 16 h (**D**). Correlation analysis was performed using Pearson’s correlation analysis.

**Figure 7 cells-11-02962-f007:**
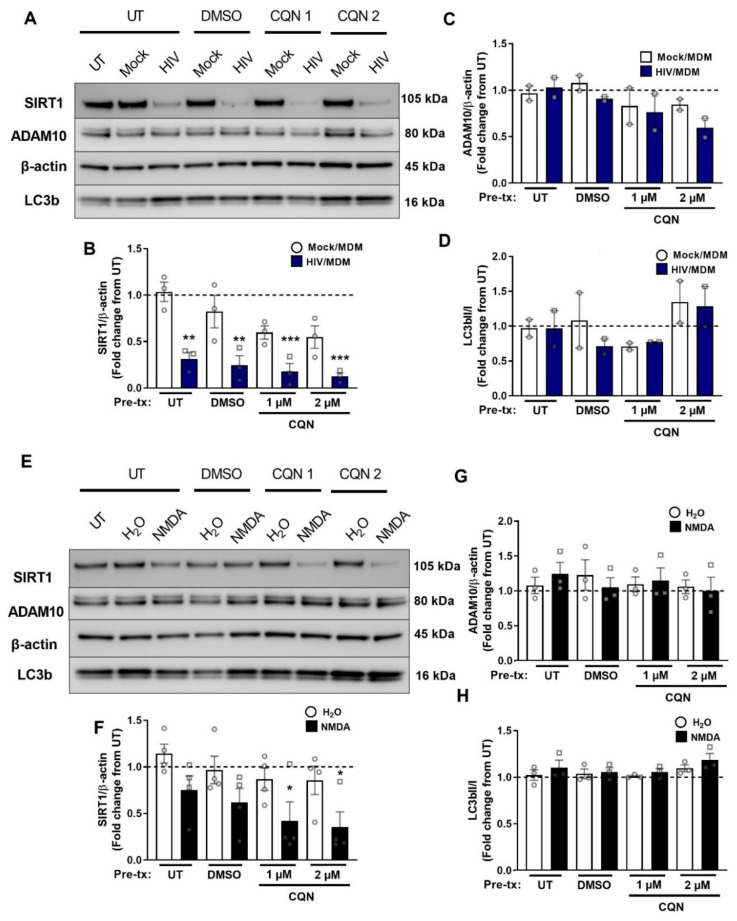
Early SIRT1 decrease by HIV/MDMs does not seem to be mediated by autophagy. Protein lysates were collected from rat neuronal cultures after a 4 h pretreatment with vehicle (DMSO) or chloroquine and a 2 h treatment with HIV/MDMs. Representative blots are shown (**A**). Chloroquine did not rescue levels of SIRT1 decrease by HIV/MDMs (**B**). Statistical significance was calculated using two-way ANOVA followed by Tukey’s post hoc. *** Represents *p* < 0.05, ** *p* < 0.01, **** p* < 0.001 compared to Mock/MDMs (*n* = 3). ADAM10 levels were unchanged (**C**) while the LC3BII/I ratio seems to tend toward an increase (**D**). Protein lysates were collected from rat neuronal cultures after a 4 h pretreatment with DMSO or CQN and a 2 h treatment with vehicle for NMDA (H_2_O) or NMDA. Representative blots are shown (**E**). Chloroquine did not seem to restore levels of SIRT1 and decreased SIRT1 (**F**). Statistical significance was calculated using two-way ANOVA followed by Tukey’s post hoc. *** Represents *p* < 0.05 compared to H_2_O (*n* = 4). ADAM10 levels seem to be unchanged (**G**) while the LC3BII/I ratio seems to tend toward an increase (**H**). Dotted line represents untreated samples.

**Figure 8 cells-11-02962-f008:**
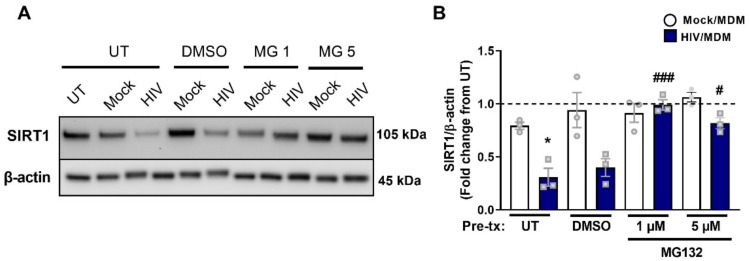
Early SIRT1 decrease by HIV/MDMs is mediated by the proteasome. Protein lysates were collected from rat neuronal cultures after a 1 h pretreatment with vehicle (DMSO) or MG132 (MG) and a 2 h treatment with HIV/MDMs. Representative blots are shown (**A**). Both doses of MG132, 1 and 5 μM, rescued the decreased SIRT1 by HIV/MDM supernatant treatment (**B**). Statistical significance was calculated using two-way ANOVA with followed by Tukey’s post hoc. *** Represents *p* < 0.05 compared to Mock/MDMs and # *p* < 0.05, *### p* < 0.005 compared to HIV/MDMs (*n* = 3). Dotted line represents untreated samples.

## Data Availability

The data presented in this study are available upon reasonable request from the corresponding author.

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
