# Peer review of "HIV-Associated Insults Modulate ADAM10 and Its Regulator Sirtuin1 in an NMDA Receptor-Dependent Manner"

_cells, 2022, doi:10.3390/cells11192962_

Round 1

Reviewer 1 Report

In this work, Lopez-LLoreda et al. studied the effect of HIV-infected human macrophages (HIV/MDMs) on the expression of ADAM10 and SIRT1 in primary cultures of rat cortical neurons. They showed that HIV/MDM treatment induced a decrease in the expression of both proteins and both decreases were blocked by the NMDA receptor antagonist MK-801. They also suggested that the decrease in SIRT1 protein levels occurs earlier than the decrease in ADAM10 protein levels. Moreover, the authors showed that SIRT1 reduction was mediated by proteasomal degradation. In this reviewer's opinion, the work is interesting, well written and well-conducted. However, there are some aspects that could be improved. Here are some suggestions:

Major points:

1.     Regarding Methods:

a.     The authors should indicate the purity of the cortical neuronal cultures obtained after treatment with Ara-C. What percentage of glial cells remains?

b.     The protocol used to differentiate monocytes into macrophages should be justify in this section.

c.      A statement on ethical issues involving animals and human samples must be included.

d.     The doses of NMDA used should be indicated in section 2.3.

e.     The authors indicated that “images were captured at 20x at 25 point per well” for the neurotoxicity studies. To know how representative is the sample, they should indicate the type/size of wells used for these experiments.

f.       To use the ANOVA test, normality of the group data and equal variances are required. Likewise, the post-hoc test is an integral part of ANOVA. This information, in addition to the significance, must be included in section 2.6.

2.     The work would benefit from demonstrating the presence of ADAM10 and SIRT1 in the cultures, as the authors based their results on WB, which depends on the specificity of the antibodies used.

3.     In the footnote of Fig. 1B, the authors included information on the meaning of *, even though this symbol is not included in the histogram. Please check the symbols (*, **, #) in all figures and clearly indicate the group to which the authors are comparing with. In Fig. 2B, 2D, 4B and 4D, are all the decreases showed statistically significant?

4.     Please include the information on the size of the bands in the WB figures.

5.   The authors indicated that 20mM NMDA induced a decrease in the expression of ADAM10 and SIRT1. Does this drug induce cell death in neuronal cortical cultures at the doses used? Does NMDA receptor blockade inhibit neuronal death induced by HIV/MDM treatment?

6.   Regarding Fig. 6, including another timepoint of analysis of ADAM10 and SIRT1 protein levels between 2 and 16 hours would be appropriate. This would make the results more robust. Moreover, the information on the method used for correlation analyses should be matched between methods and figure 6.

Minor point:

1.     The meaning of the abbreviation SIVE should be indicated.

Author Response

We greatly appreciate the feedback from the reviewers and appreciate the opportunity to respond to their constructive comments.  Please see attached file for responses. 

Reviewer 2 Report

The aim of this study is to investigate whether HIV-associated insults modulate the expression levels of the non-amyloidogenic enzyme ADAM10 and its regulator SIRT1. The manuscript is well written, the figures are well presented, the results are well explained, and the analysis was performed using appropriate statistical methods. However, there are some concerns that I highly recommend addressing:

1-       The use of cells from different species (rat cortical cultures and human monocytes) decreases the relevance of the paper.

2-          Several studies have shown that pericytes can be infected by HIV. Please include pericytes with the other BBB cells mentioned on the introduction.

3-    How were HIV levels measured from the supernatant of infected macrophages?

4-          There is no explanation on material and methods of how HIV Jago, used for the experiments, was produce and quantified.

5-           An explanation of why the authors decided to use HIV jago strains would be interesting given that several HIV strains as NL-43 have shown to be more effective in terms of HIV infection on cells of the BBB.

6-           In figure 1. Please complement the figure with new results of cell viability.

7-       In figure 7, it will be important to show the graphical results not only of SIRT1 but also of L3Cb or ADAM10 levels. Importantly, LC3bII levels analysis should be performed comparing the levels of L3CbII between the levels of L3CbI. Moreover, measuring the levels of LC3BII is not enough evidence to assert the result 3.7. Please include levels of autophagy markers such as p62 or others and supplement the results with visual detection methods, such as immunofluorescence.

Author Response

(The authors gave the same response as above.)

Reviewer 3 Report

The purpose of the current paper is to determine whether HIV/MDM can affect the expression levels of ADAM10 and Sirtuin1 (SIRT1) in neurons -  ADAM10 known to be implicated in ageing and in AD context.

This is an interesting study, performed in vitro using primary rat neurons culture treated or not with HIV/macrophages supernatant.

Several points need to be clarified (see below).

Material and methods: 

- line 130: ... neurons were treated with Mock/MDMs or HIV/MDMs.

The authors need to specify in the material and methods section as well as in each corresponding figures the ratio of monocyte supernatant per neuron (1 macrophage :1 neuron, or per amount of protein secreted by macrophages for a constant number of neurons to ensure a similar treatment between Mock/MDMD and HIV/MDM). In other word the authors should show that they  used the same parameters to treat neurons with either Mock and HIV supernatants, and that this treatment is physiologically relevant.

- Figure 2:  What the number 1,2 3,4 and 5 represent in Fig2D. It seems that Fig2 A and B showing the variability across primary neuron culture.  Then what are the conditions for fig 2C and D: same cell culture isolation treated with 5 different HIV/macrophages supernatants? Can the authors clarify? If it is correct, then may I suggest to improve clarity for the paper to put in supplemental data the  variability across independent culture preparation (Fig2 A and B), and only highlight in the main text the main results (Fig2 C and D). Similar comments for figure 4.

- Figure 6: The authors are claiming that SIRT1 levels changes before ADAM10. To confirm this statement, a time course experiment should be added to show (1) in their experimental conditions ADAM will change and (2) that the changes in SIRT1 and ADAM 10 are happening at different time.

Regarding the correlation in figure 6D, the authors should make clear that this correlation is observed in in HIV/MDM treated samples, as it is confusing at the moment. Indeed NMDA treatment is used for Fig6A-C.

- Figure 8: To close the loop regarding the involvement of the proteasome pathway in SIRT1 degradation, would it be possible to confirm that the proteasome activity and/or level of ubiquitinylated SIRT1 is increased in HIV/MDM treated cells?

Minor:

- line 353: "/" missing in "HIVMDM"

Author Response

(The authors gave the same response as above.)

Round 2

Reviewer 1 Report

In the revised version of the paper the authors have properly considered the reviewer’s comments and have incorporated the suggested modifications with the corresponding explanations. I would simply recommend including a sentence in the text stating that the presence of ADAM10 and SIRT1 has been confirmed in the cultures using PCR and include the image of the gel as supplementary material.

Author Response

Thank you for your comments, we are glad to hear that you feel we addressed everything appropriately. We have added a sentence in the text mentioning that we confirmed the presence of ADAM10 and SIRT1 and have included the gel as Figure 2 of the supplementary data. Again, thank you for your help in improving this manuscript. 

Reviewer 2 Report

I agree with the changes made by the authors.

Author Response

We are glad to hear that you feel we addressed everything appropriately. Thank you for your comments and for your help in improving this manuscript. 

Reviewer 3 Report

The authors answered all my comments.

In Supplemental figure 1: a legend for the sub-figure c is missing.

Author Response

We are glad to hear that you feel we addressed everything appropriately. Thank you for pointing this out, we have added the information for the caption of Suppl. Figure 1. Thank you for your comments and for your help in improving this manuscript.